# Patterns of T and B cell responses to *Mycobacterium tuberculosis* membrane-associated antigens and their relationship with disease activity in rheumatoid arthritis patients with latent tuberculosis infection

Shashi Kant Kumar, Suvrat Arya, Ankita Singh, Ramnath Misra, Amita Aggarwal*, Sudhir Sinha ⓘ *

Department of Clinical Immunology & Rheumatology, Sanjay Gandhi Postgraduate Institute of Medical Sciences (SGPGIMS), Lucknow, India

* ssinha@sgpgi.ac.in, sinha.sudhir@gmail.com (SS); amita@sgpgi.ac.in, aa.amita@gmail.com (AA)

## Abstract

This study was aimed at exploring whether latent tuberculosis infection (LTBI) contributes to the pathogenesis of immune-mediated inflammatory diseases in a TB endemic setting. We screened 198 rheumatoid arthritis (RA) patients with tuberculin skin test (TST) and studied 61 (median DAS28-ESR = 6.3) who were positive. Whole blood T cell proliferative responses to *Mycobacterium tuberculosis* (Mtb) membrane (MtM) antigens, including the latency-induced protein alpha crystallin (Acr), were determined by flow cytometry using Ki67 expression as the marker for nuclear proliferation. Serum antibody levels were determined by ELISA. Follow-up investigations (at 3–6, 9–12 and 15–18 months after baseline) were performed in 41 patients who were classified empirically as 'high' (HR-T/HR-B) or 'low' (LR-T/LR-B) responders based on their dynamic T cell or antibody responses. Significant correlations were seen between baseline T cell responses to MtM and Acr, and between IgG, IgA and IgM antibody responses to MtM. However, no correlation was seen between T and B cell responses. At all time points during the follow-up, T cell responses to both antigens (except for MtM at one point) were significantly higher in HR-T (n = 25) than LR-T (n = 16) patients. Levels of IgA and IgM (but not IgG) antibodies to MtM were also significantly higher in HR-B (n = 13) than LR-B (n = 28) at all time points. Importantly, HR-T patients exhibited significantly higher baseline and follow-up DAS28 scores than LR-T. Ten (of 61) patients had a history of TB and developed RA 6 years (median) after contracting TB. Three new TB cases (1 from TST-positive and 2 from TST-negative groups) emerged during the follow-up. Our results suggest that persistently elevated T cell responses to Mtb antigens may contribute to disease activity in RA.

**Data Availability Statement:** All relevant data are within the manuscript and its Supporting Information files.

**Funding:** The authors received no specific funding for this work.

**Competing interests:** The authors have declared that no competing interests exist.

## Introduction

Rheumatoid arthritis (RA) is an immune-mediated inflammatory disease (IMID) caused by interplay of genetic and environmental factors that dysregulate the immune system [1]. Treatment with disease modifying anti-rheumatic drugs (DMARDs) acting as tumour necrosis factor-alpha (TNF) blockers makes IMID patients susceptible to TB [2] due mainly to reactivation of latent TB infection (LTBI), defined as 'a state of persistent immune responsiveness to *Mycobacterium tuberculosis* (Mtb) antigens without clinically manifested TB' [3]. In some population-based studies [4, 5] history of TB was found associated with RA suggesting that Mtb could also play a role in pathogenesis of RA which, like TB, may begin in the airways and lungs [6]. In early RA disease, germinal centre-like structures are seen in the lungs wherein activated B cells produce antibodies to citrullinated protein antigens (ACPAs). Seropositive RA patients harbour citrullinated proteins in their bronchial tissue and this abnormality was not associated with smoking [7, 8]. Pulmonary TB patients also show seropositivity for ACPA and rheumatoid factor (RF) [9] suggesting that Mtb could trigger the production of these antibodies which are otherwise considered as hallmarks of RA.

Several studies have indicated that immune-inflammatory responses to Mtb, as well as non-tuberculous mycobacteria (NTM), can drive the pathogenesis of IMIDs and signaling via toll-like receptors (TLRs) plays an important role in this process [10, 11]. Cell membrane-associated glycolipids and lipoproteins of Mtb serve as microbe-associated molecular patterns (MAMPs) recognized by the pattern-recognition receptors (PRRs, including TLRs) of innate immune cells [11]. The pro-inflammatory cytokines produced by these cells, besides orchestrating innate immunity against the invading microbe, can trigger pathogenesis of RA and other IMIDs [6]. Antigenic constituents of Mtb membrane (MtM) are also potent inducers of adaptive immunity which, in turn, may contribute to the pool of pro-inflammatory cytokines [10, 12]. Mtb 'immunoproteome' is rich in membrane-associated and secreted proteins [13] and certain MtM-associated proteins identified by us [14] and others [15] were found capable of inducing strong T cell responses in persons with LTBI. In addition, MtM antigens are also strong inducers of B cell responses leading to production of high titres of antibodies [16–18]. The MtM-associated stress protein alpha crystallin (Acr, Rv2031c) sustains Mtb during its latent phase [19]. Its overproduction by the dormant bacilli and occurrence within 'Mtb complex' make Acr an important latency-associated antigen [20]. As shown by us [17] and others [21], it induces strong T- and B cell responses in persons with LTBI. Acr was also shown to induce T cell responses in RA patients harbouring LTBI [22].

Immune responses to Mtb are dynamic in nature. T cell responses vary according to the bacterial load and decline with treatment [23, 24]. Specificity and isotypes of antibodies produced by B cells also vary according to the state of infection. While persons with LTBI preferentially produce antibodies to MtM-associated antigens, those with active TB do so against 'secreted' antigen of Mtb [13, 25]. In a longitudinal study [26], TB patients showed higher IgA levels against Acr than subjects with LTBI. In another study [27], consistently elevated levels of IgA against Mtb antigens were considered a risk factor for progression to active TB disease. However, despite the apparent similarities in their dynamic trends, T and B cell responses to Mtb may not go hand in hand. A cohort of household contacts from TB endemic area was shown to harbour antibodies to Acr and other MtM-associated antigens while remaining negative for T cell responses determined by tuberculin skin test (TST) or interferon gamma release assay (IGRA) [28].

India is home to over a quarter of TB patients diagnosed worldwide [29] and majority of apparently healthy Indians harbour LTBI [30, 31]. Further, the dynamic nature of host's immunity is believed to determine whether LTBI will progress to active TB disease or regress

to a 'self-limiting' state [32]. Against this backdrop, and the indications that Mtb could serve as an environmental trigger for IMIDs, we undertook this study aimed at exploring whether the RA patients harbouring LTBI exhibit distinct T and B cell response patterns against MtM-associated antigens and whether such patterns have any relationship with their disease activity.

## Materials and methods

### Ethics statement

The study protocol was approved by Institutional Ethics Committee of SGPGIMS (IEC code: 2016-149-IMP-EXP). All participants provided a written informed consent.

### Study group

One hundred ninety eight RA patients attending the Rheumatology clinic at our hospital were examined for fulfilment of the following inclusion criteria: Adults with active disease [33], taking ≤10 mg prednisolone/day, no sign or symptom of TB or any other co-morbidity, and a positive TST. Sixty one (30.8%) patients met these criteria and were recruited for the study.

### Tuberculin skin test

TST was performed as per recommendations of IUATLD and WHO [34]. Accordingly, 5 tuberculin units (TU) of PPD (Arkray Healthcare, Surat, India) were injected intradermally on the left forearm. After 48–72 hours, mean diameter of induration was measured using a calliper. TST was considered as positive if the induration was 10 mm or more.

### Blood samples

At the baseline and follow-up intervals, 1 ml venous blood was collected in heparin tubes (for T cell assays) and 3 ml was collected in plain tubes (to get sera for antibody assays).

### Mtb antigens

Mtb cell-membrane (MtM) was isolated by using a previously described protocol [17]. Briefly, 3–4 week old culture of Mtb (strain H37Ra, ATCC25177) was harvested and sonicated. Cell lysate was differentially centrifuged to obtain membrane as sediment and cytosol as supernatant. Protein was estimated by the modified Lowry's method [17] and aliquots were stored at -80°C. Identity of Mtb was ascertained by immune-chromatographic detection of MPT64 [17] using a kit ('SD Bioline TB Ag MPT64 Rapid', Abbott, USA). Recombinant Acr protein (LRP-0019C) was purchased from Lionex GmbH, Germany.

### T cell proliferation assay

A previously described protocol [17] was used with some modifications. In brief, blood samples (diluted 1:10) were dispensed (1 ml/well) in 24-well culture plates and incubated with test antigens (MtM or Acr, 5μg/ml) or controls (culture medium as negative and PHA as positive control) for 5 days in a $CO_2$ incubator. Harvested cells were stained with fluorescent anti-CD3 antibody and RBCs were lysed. Leukocytes were fixed with 2% paraformaldehyde and permeabilised with 0.2% Triton-X100. Washed cells were stained with fluorescent anti-Ki67 antibody. Cells were finally washed and suspended in PBS. Data on $10^5$ cells in lymphocyte gate was acquired on a flow cytometer (BD FACS Canto-II) and analysed with FlowJo software (Tree Star). Percent responder cells (CD3+Ki67+) for antigens or mitogen (PHA) were determined

by subtracting corresponding values for culture medium. Gating strategy is shown in S1 Fig in S1 File and representative flow plots are shown in S2 Fig in S1 File.

## ELISA

A previously described protocol [17] was used. In brief, ELISA plates were coated with antigens (10 µg/ml MtM or 1 µg/ml Acr) or coating-buffer and blocked with 2% skimmed milk powder in tris-buffered saline containing 0·05% Tween 20 (TBS-T). Diluted test sera (1:500 for MtM or 1:100 for Acr) were dispensed in antigen and buffer-coated wells and incubated. Washed plates were re-incubated with peroxidase-conjugated antibodies to human IgG, IgA or IgM. Plates were finally washed and incubated with the substrate (o-phenylene diamine). Reaction was stopped with 7% $H_2SO_4$ and optical densities (ODs) were read at 492 nm. For each serum, mean OD with buffer was subtracted from mean OD with antigen and expressed as ΔOD.

## Classification of patients as 'high' or 'low' responders for Mtb antigens

Forty one patients (out of 61 recruited) returned for at least one of the 3 follow-up investigations scheduled at 3–6, 9–12 and 15–18 months after baseline. They were classified as 'high' or 'low' responders for T cell or antibody responses based on the following empirical criteria: For T cell responses, patients showing >2 fold increase (from baseline) in response to either or both antigens (MtM/Acr), at one or more time points, were classified as high responders (HR-T). Conversely, low responders (LR-T) were those who showed ≤2 fold increase in response to both the antigens at all time points. For antibody responses, patients showing >2 fold increase in the level of any antibody isotype (IgG, IgA or IgM), at one or more time points, were classified as high responders (HR-B). Low responders (LR-B) were those who showed ≤2 fold increase in all antibody isotypes at all time points.

## Statistical analysis

Non-parametric statistical methods were applied. Correlation between two variables was assessed by Spearman's rho. Difference between two proportions was assessed by Fisher's exact test. Differences between paired datasets were assessed by Wilcoxon test and those between unpaired datasets were assessed by Mann-Whitney test. P values <0.05 were considered as significant. All calculations were performed using Graphpad Prism software.

## Results

### Baseline characteristics of study subjects

Most (78.7%) of the 61 TST-positive RA patients enrolled for the study were females and about half of them were on low-dose (≤10 mg/day) prednisolone (Table 1).

**Table 1. Baseline characteristics of the study subjects.**

| S. No. | Variables | Values |
|---|---|---|
| 1. | Number of patients (females) | 61 (48) |
| 2. | Median age in years (inter-quartile range or IQR) | 41 (31–50) |
| 3. | Number with RF/ACPA | 49/46 |
| 4. | Median disease duration in years (IQR) | 4 (1.5–6) |
| 5. | Median DAS28-ESR (IQR) | 6.3 (5.6–7.2) |
| 6. | Number using ≤10 mg prednisolone/day | 34 |
| 7. | Median TST induration in mm (IQR) | 13 (12–18) |
| 8. | Number with BCG vaccination scar | 34 |
| 9. | Number with history of cured TB | 10 |

## TST responses were not affected by BCG

Out of 198 patients who were screened for inclusion in this study, BCG scar was present in 55.7% (34/61) of TST-positive and 45.3% (62/137) of TST-negative patients. This difference was not statistically significant (P = 0.2179) suggesting that BCG vaccination at birth (as practised in India) did not affect TST results in the adults.

## MtM induced higher T and B cell responses than Acr

Baseline T cell proliferative responses to PHA (median, IQR; 72, 40 to 86), MtM (0.78, 0.45 to 1.8) and Acr (0.33, 0.12 to 0.72) in 58 study subjects (samples from 3 had technical issue) are shown in Fig 1A. The responses to MtM were significantly higher than to Acr. Even so, a strong correlation (r = 0.69) was seen between the two responses (Fig 1B) suggesting that MtM could provide a more sensitive alternative to Acr for monitoring the T cell responses to Mtb.

Baseline antibody responses to MtM and Acr are shown in Fig 2A. Anti-MtM IgG levels (median ΔOD, IQR; 0.28, 0.14 to 0.5) were significantly higher than IgA (0.11, 0.08 to 0.2) and IgM (0.2, 0.1 to 0.33) and IgM levels were also significantly higher than IgA. Despite these differences, significant correlations were seen between all 3 isotypes (Fig 2B–2D) indicating that they could broadly be targeting the same set of antigens. Acr did not elicit any appreciable antibody response (0, -0.01 to 0.03 for IgG; 0, -0.02 to 0.02 for IgA and -0.01, -0.04 to -0.54 for IgM) which suggested that MtM could work as a more sensitive alternative to Acr for monitoring of anti-Mtb antibody responses also. In view of these results, further analyses of antibody responses were based on anti-MtM antibodies alone.

We did not observe any correlation between baseline T cell and antibody responses against MtM (S3 Fig in S1 File) suggesting that the antigenic determinants eliciting these responses could, at least partly, be non-overlapping.

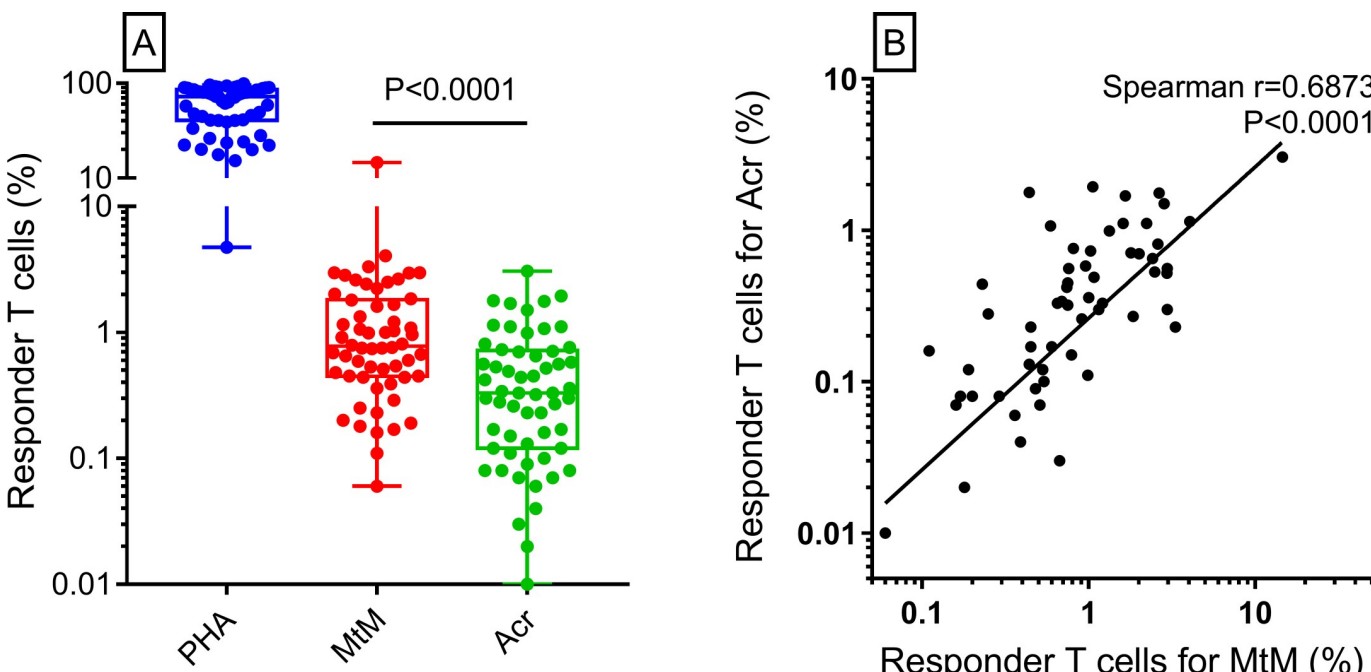

**Fig 1. Baseline T cell proliferative responses of the study subjects (n = 58).** Panel A shows percent responder T cells (CD3+Ki67+) against the mitogen (PHA) and Mtb antigens (MtM and Acr). Though the responses to Acr were significantly lower than to MtM, a significant correlation was seen between the two (Panel B). P (Mann-Whitney) and r (Spearman's) values are shown in the panels.

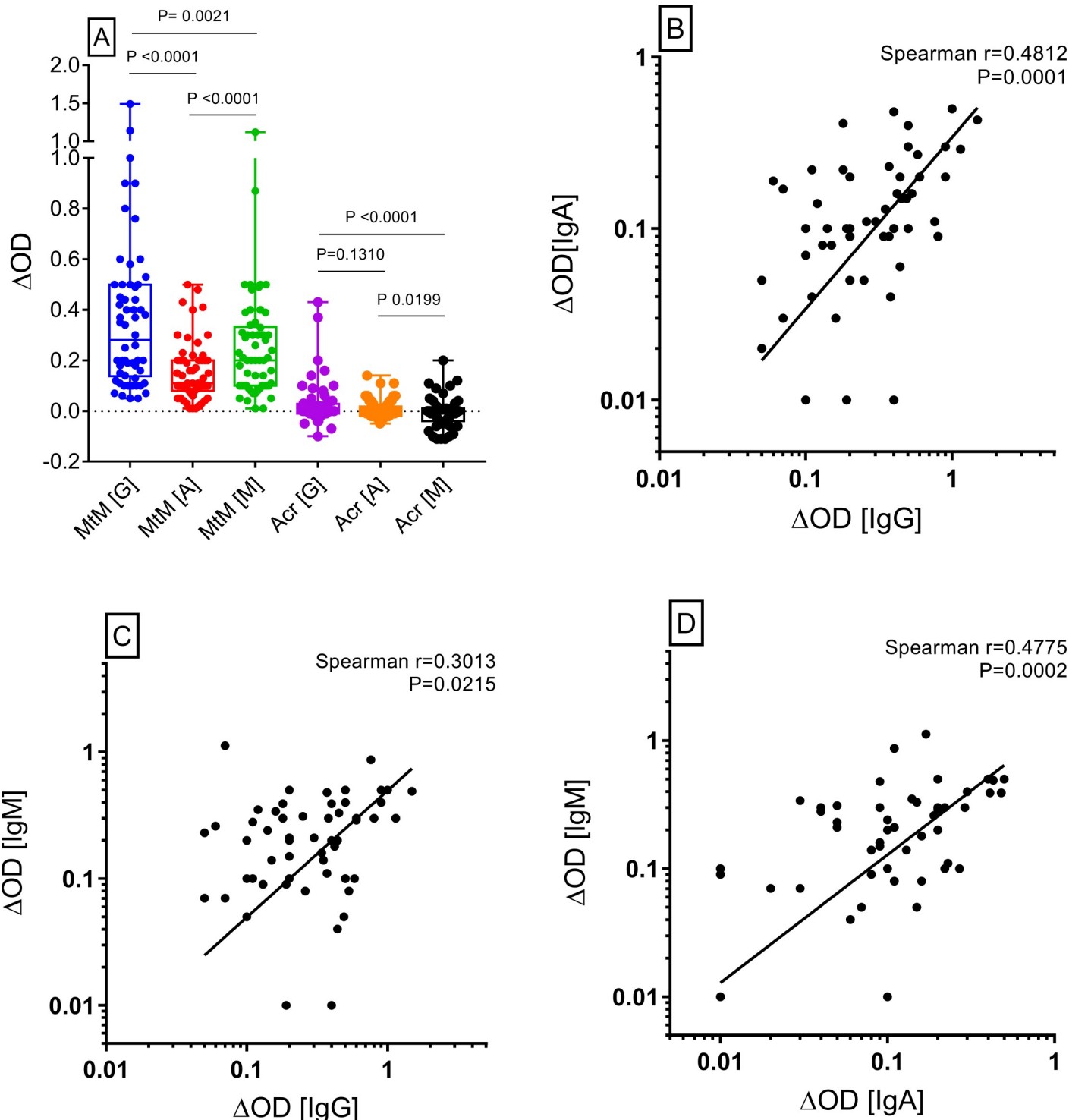

**Fig 2. Baseline B cell responses of the study subjects (n = 58).** Levels (ΔOD = mean OD of antigen-coated minus mean OD of buffer-coated wells) of anti-MtM IgG antibodies were highest followed by IgM and IgA (Panel A). Despite the significant differences, levels of all 3 antibody isotypes correlated with each other (Panels B-D). Acr did not produce any appreciable antibody response. P (Mann-Whitney) and r (Spearman's) values are shown in the panels.

**Patients showed distinct immune response profiles against Mtb antigens** According to the applied criteria (see Materials and Methods), 25 patients (out of 41 who were followed-up) were categorised as 'high' responders (HR-T) and the remaining 16 as 'low' responders (LR-T) for T cell responses to the Mtb antigens (MtM and Acr). Likewise, 13 patients were high responders (HR-B) and 28 were low responders (LR-B) for B cell (antibody) responses to MtM. Consistent with the observed lack of correlation between baseline T and B cell responses, no association (P = 0.5134) was seen between HR-T and HR-B, or LR-T and LR-B patient categories (S1 Table in S1 File).

Sequential T cell responses ($\log_2$ fold-change from baseline responses) of HR-T and LR-T patients against MtM and Acr are displayed in Fig 3A. Both antigens showed comparable response profiles, consistent with the observed correlation between two responses at the baseline. At all 3 time points of follow-up, responses to both the antigens (except for MtM at 15–18 months) in HR-T were significantly higher than LR-T patients. Interestingly, a progressively rising trend was seen in the responses, particularly of LR-T patients who showed a significant rise in response to Acr between 3–6 and 15–18 months.

Sequential antibody responses to MtM (Fig 3B) were distinctly different from the corresponding T cell responses. A significantly higher (P = 0.0143) proportion of patients fell under the LR-B (68%) compared with LR-T (39%) category, suggesting that RA patients with LTBI were able to mount relatively weak antibody responses. Moreover, unlike T cell responses, antibody levels ($\log_2$ fold-change from the baseline) of HR-B as well as LR-B patients remained almost stable throughout the follow-up period. Interestingly, though statistically significant differences were seen in IgA and IgM levels of HR-B and LR-B patients at all time points, IgG levels did not differ at any point.

Altogether, these results suggest that the RA patients with LTBI may fall into distinct categories with respect to their longitudinal immune responses against Mtb antigens. They also underscore the role played by immunoglobulin isotypes in shaping the overall antibody response.

## Prescribed treatment did not affect the T or B cell response profiles

During the follow-up, patients were treated with prednisolone (up to 10 mg/day) along with methotrexate (up to 25 mg/week). In addition, some patients also received hydroxychloroquine (up to 400 mg/day). The proportions of HR and LR patients (for T as well as B cell responses) receiving these drugs were comparable (S4 Fig in S1 File), suggesting that the administered treatment did not have any remarkable effect on the observed immune response profiles.

## Patients with persistently high T cell responses to Mtb antigens also showed high disease activity

DAS28-ESR scores of HR-T and LR-T patients at the baseline and at two follow-up intervals are shown in Fig 4A. As expected, disease activity in all patients declined with treatment. However, at all time points, disease activity scores in HR-T remained significantly higher than LR-T patients. In addition, the duration of disease was significantly shorter in HR-T patients (Fig 4B). These results suggested that RA patients with persistently high T cell response to Mtb antigens could also exhibit high disease activity which could be attained shortly after disease onset.

Unlike T cell responses, antibody responses did not show any association with disease activity or duration (Fig 4C, 4D) suggesting that antibodies to Mtb antigens may have a relatively lesser relevance for the pathogenesis of RA.

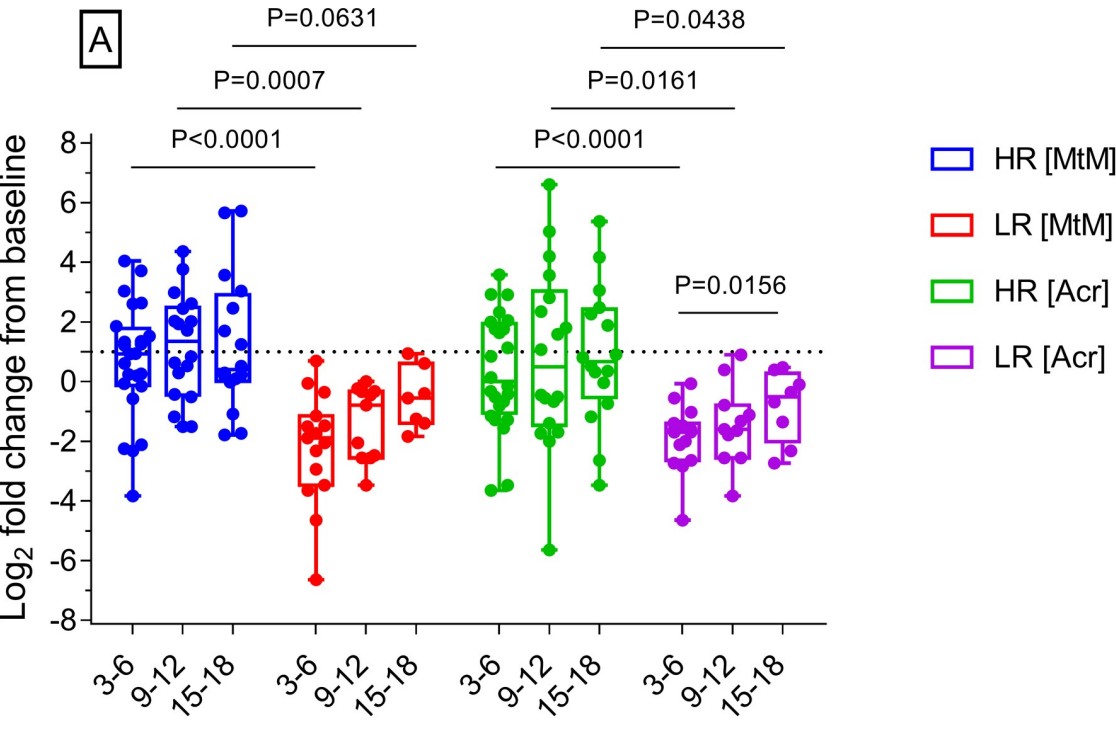

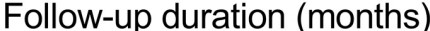

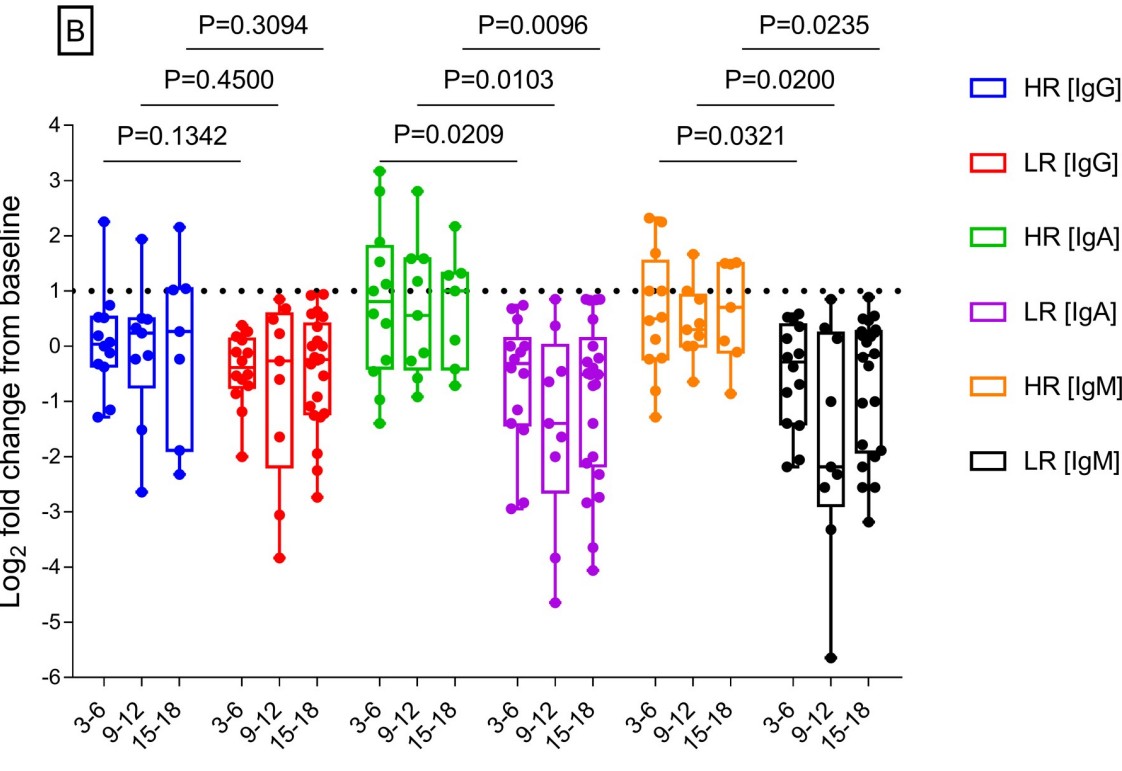

**Fig 3. Sequential T and B cell responses of the study subjects (n = 41).** Dotted lines denote two-fold ($\log_2 = 1$) higher response from the baseline (0). Significant differences between high (HR-T, n = 25) and low (LR-T, n = 16) T cell responders were seen for Acr at all 3 and for MtM at 2 follow-up intervals. Responses in LR-T were mostly below baseline but showed a rising trend which attained statistical significance in case of Acr (Panel A). Significant differences were also seen between high (HR-B, n = 13) and low (LR-B, n = 28) B cell responders, for anti-MtM IgA and IgM (but not IgG) antibodies at all time points (Panel B). P values for observed differences (Mann-Whitney test for inter-group and Wilcoxon test for intra-group comparisons) are shown on top of the selected columns.

## Association between TB and RA

Out of 61 patients recruited for the study, 10 (16.4%) had a history of cured TB, of whom 7 had suffered from pulmonary TB. The median time taken for the onset of clinically active RA after contracting TB was 6 years (Table 2).

All 198 patients (82% females) who were screened for inclusion in the study were monitored for development of TB over the study period. One new TB case emerged in the TST-positive (n = 61) and 2 new cases emerged in TST-negative (n = 137) patient groups. Thus the proportions of incident cases in both groups were comparable.

## Discussion

In absence of a 'gold standard' method for detection of LTBI, we chose TST for several reasons. As comparisons of TST and IGRA in the same population have not provided evidence that one test is superior to the other, WHO has recommended that both are equivalent options, with TST being the method of choice in resource-constrained settings [3]. The choice of 10 mm reaction as cut-off for positivity was aimed at detecting those patients who may have a greater propensity to develop TB. In a large TST trial from south India [35], reactions >10 mm showed a higher association with TB development than those <10 mm. A TBNET study [36] has also shown that TST cut-off of 10 mm is most appropriate for the diagnosis of LTBI in RA patients since loss in sensitivity by increasing the cut-off from 5 to 10 mm is marginal while there is substantial gain in specificity. The rate of TST positivity in our patients (30.8%) was comparable to that (30%) in another study on RA patients from India, despite the authors choosing to use a double strength (10 TU) of PPD [37].

BCG vaccination at birth (as recommended by WHO and practised in India) did not affect the TST responses, which is consistent with another study on RA patients [38] and our earlier study on health care workers (HCWs) [31]. Apprehensions that TST could be confounded by BCG have not found support in a meta-analysis of 24 studies comprising 240,203 subjects [39]. The study concluded that BCG given in infancy had minimal effect on TST, particularly after 10 years of vaccination. The same meta-analysis also included 18 studies comprising 1,169,105 subjects to conclude that false positive TST due to NTM infection was minimal (up to 2.3%) and reached the upper limit only in areas of high NTM and low TB prevalence (e.g., USA and Europe). This and other similar observations have prompted WHO to recommend that BCG vaccination should not be a determining factor in selecting a test (TST or IGRA) [3]. Nearly half (55.7%) of our study subjects were on low-dose (≤10 mg) prednisolone suggesting that their treatment status did not affect TST responses. Several prior studies have also shown that corticosteroids (up to 15 mg/day) or DMARDs do not affect TST responses [38, 40, 41].

Baseline T and B cell responses to MtM were significantly higher than to Acr which was expected since MtM comprises, besides Acr, several putative antigens of Mtb [14, 16]. We had also shown that MtM induces comparable and apparently mycobacterium-specific CD4+ and CD8+ T cell responses [14, 31]. Despite the difference in intensities, a strong correlation was seen between T cell responses to MtM and Acr, similar to that reported by us in HCWs [17]. The serum IgG levels against MtM were significantly higher than IgA and IgM. Even so, levels

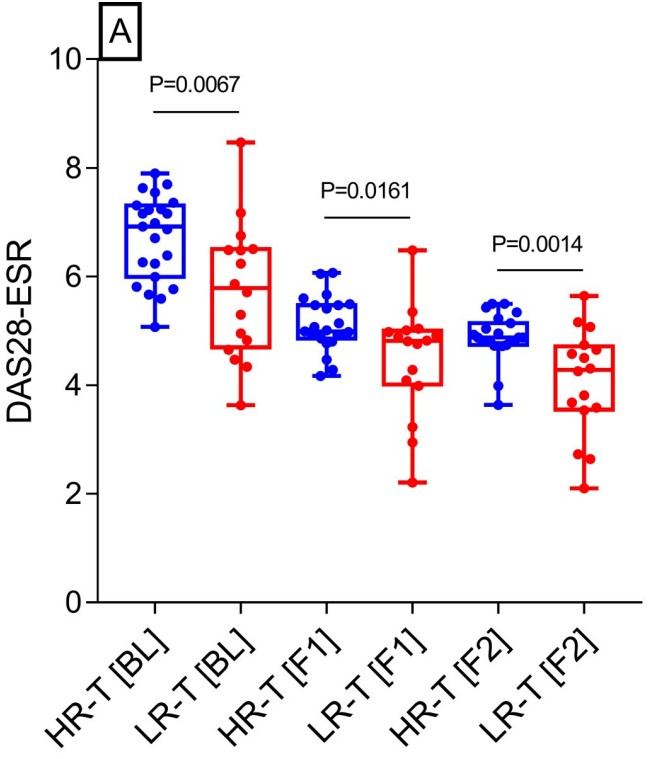

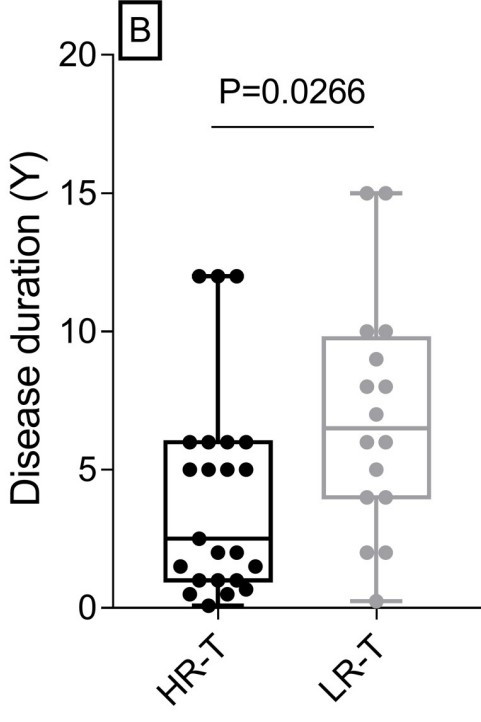

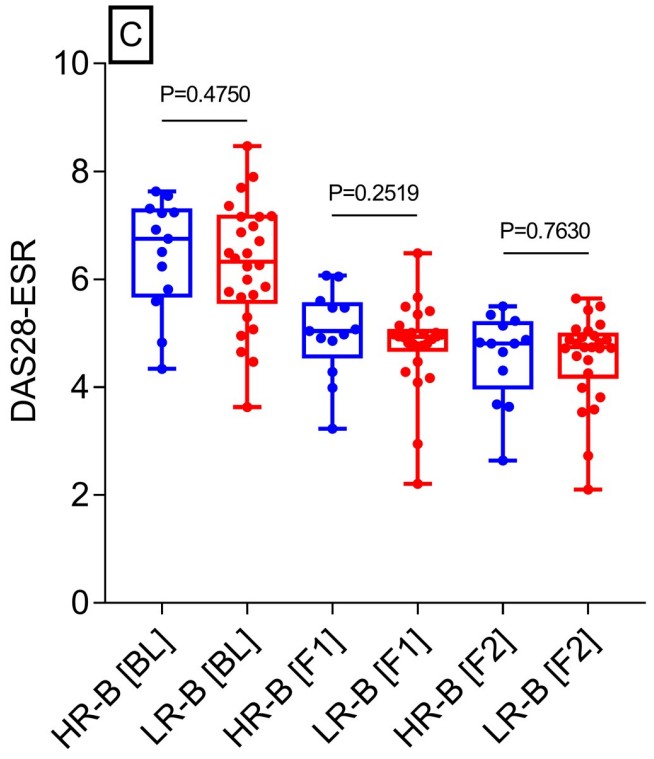

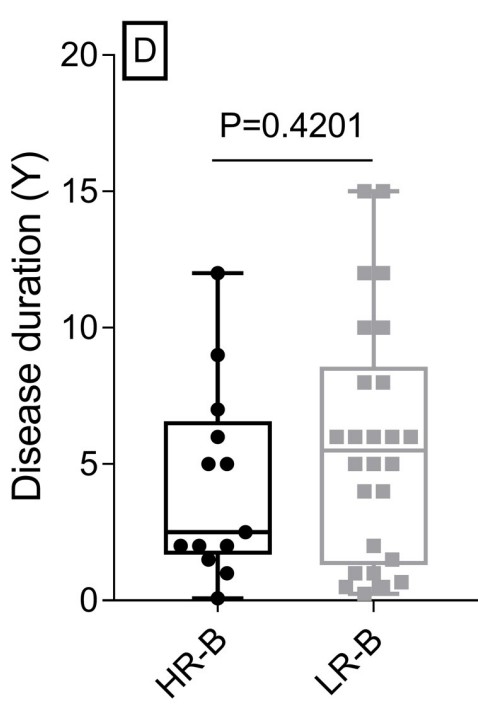

**Fig 4. Association of RA disease activity with T cell responses to Mtb antigens.** Disease activity scores (DAS28-ESR) of patients who were high responders for T cells (HR-T) were significantly higher than those of low responders (LR-T) at the baseline (BL) and also at 2 follow-up intervals (F1 and F2, Panel A). Additionally, disease duration (years) was significantly shorter in HR-T than LR-T patients (Panel B). DAS28-ESR scores of high and low responders for B cells (HR-B and LR-B) at the baseline as well as follow-up intervals did not show any significant difference (Panel C). Disease duration also showed no difference (Panel D). P values (Mann-Whitney) are shown on top of the selected columns.

of all 3 antibody isotypes correlated with each other indicating that they could broadly be targeting the same set of antigens. Contrary to responses induced by it in HCWs and TB patients [17, 26], Acr did not induce any remarkable antibody response in RA patients owing, most probably, to their dysregulated immune system [1]. Overall, these results indicated that MtM could serve as a more sensitive alternative to Acr for monitoring the T as well as B cell responses to Mtb [17]. Lack of correlation between T and B cell responses was another notable observation suggesting that they could be directed at non-overlapping antigenic determinants. This discordance was also flagged in a recent study [28] wherein a group of household contacts who were negative for TST and IGRA had shown antibodies to MtM-associated antigens. In related studies [42, 43], RA patients had also shown antibodies to mycobacterial proteins PknG, PtpA and MAP_4027. Though the authors had sourced these proteins from *Mycobacterium avium* subsp. paratuberculosis (MAP), they are also present in Mtb.

The patients classified as 'high' T cell responders (HR-T) showed significantly higher responses to MtM and Acr than 'low' responders (LR-T) at all 3 time points (except for MtM at one point) of the follow-up. Since T cell responses to Mtb decline with reduction in bacterial load [23, 24], HR-T patients could be considered as having a higher load of infection. A progressive rise in T cell responses, as seen by us, could further indicate an ongoing or persistent infection [32]. Sequential antibody (IgA and IgM, though not IgG) responses to MtM were also significantly higher in HR-B than LR-B patients at all time points. Antibodies to Mtb may perpetuate LTBI by restricting the spread of infection [16, 18]. The respiratory mucosa contains an abundance of IgA and a lower level of IgM, both of which can protect mucosal epithelium from invasion by Mtb. In RA patients also, a prominent IgA response precedes the joint inflammation [6]. Unlike the rising trends seen in T cell responses, antibody responses in either category of patients remained almost stable throughout the follow-up, suggesting that they may have reached a plateau even before baseline investigations. This finding resonates with the fact that RA-associated antibodies can be present in blood several years prior to

**Table 2. Association between TB and RA.**

| Patient No.[a] | Age/Sex | BCG | TST (mm) | TB Type | Time since TB cure (Years) | Time-lag between TB cure and RA onset (Years) |
|---|---|---|---|---|---|---|
| 1 | 46/F | 1[b] | 12 | Lymphnodal | 22 | 7 |
| 2 | 60/F | 0 | 10 | Pulmonary | 15 | 13 |
| 3 | 25/F | 1 | 19 | Pulmonary | 10 | 6 |
| 4 | 36/F | 1 | 18 | Endometrial | 6 | 2 |
| 5 | 58/M | 0 | 20 | Pulmonary | 24 | 15 |
| 6 | 49/F | 1 | 16 | Pulmonary | Not known | Not known |
| 7 | 22/F | 0 | 12 | Pleural | 6 | 4 |
| 8 | 49/F | 1 | 18 | Pulmonary | 15 | 4 |
| 9 | 37/F | 0 | 19 | Pulmonary | 14 | 2 |
| 10 | 32/F | 1 | 15 | Pulmonary | 14 | 9 |
| | | | | Median (IQR) | 14 (8–19) | 6 (3–11) |

[a]Nine patients (except No. 4) were positive for RF. Seven patients (except Nos.1, 2 and 4) were positive for ACPA. [b]1 = BCG scar-positive, 0 = scar-negative.

disease onset [1, 6]. Compared with HR-T, significantly lesser proportion of patients fell in the HR-B category, suggesting that their B cell responses may be relatively subdued. No associations were seen between HR-T and HR-B or LR-T and LR-B categories of patients which reflected the lack of correlation between T and B cell responses at the baseline. We did also not observe any effect of treatment on the T or B cell responses which is consistent with the reports showing that corticosteroids and DMARDs, at the doses administered by us, did not affect TST responses [38, 40, 41].

An important observation of this study was significantly higher disease activity scores (DAS28) in HR-T than LR-T patients at baseline as well as during the follow-up. RA patients characteristically express HLA-DR4 [44] which points to a critical role of T cell responses in disease development. Analysis of T cell responses in RA has mostly remained focused on auto-antigens identified by autoantibodies and candidate peptides for the selected autoantigens (e.g., citrullinated proteins) are defined by their binding to HLA-DR alleles associated with seropositive RA [45]. However, a major limitation of this approach is its inability to look for other antigens which could cause T cell activation particularly when one third of RA patients are seronegative [6]. A major study has shown that T cell responses to Mtb antigens are also executed through HLA-DR4 and suggested that this could drive the pathogenesis of RA [46]. In addition, cloned T cells from synovial fluid of RA patients have shown reactivity to Mtb antigens [47]. The antibody responses against Mtb, on the other hand, did not show any association with disease activity. In an earlier study also, antibodies to the Mtb antigens PknG and PtpA did not show such an association [42] suggesting that the anti-Mtb antibodies may not play a significant role in RA pathogenesis.

The fact that 16% of our enrolled patients had a history of TB also points to Mtb as a possible 'environmental' factor for RA. A recent study from Taiwan has found evidence of past TB, detected by chest x-ray, in a similar proportion of IMID patients, mostly of RA [48]. In another large Taiwanese cohort, 2.38% RA patients presented a history of TB whereas TB prevalence in the control population was 1.36% [5]. Our patients developed RA 6 years (median) after contracting TB whereas in Taiwanese cohort [5] the median time to develop RA was 3.61 years. These differences could be reconciled with the differences in cohort sizes or demography.

We followed all 198 patients who were screened for selection of study subjects for the development of TB. Three new TB cases (1 in TST-positive and 2 in TST-negative patients) emerged during the study period. As lifetime risk of conversion of LTBI to active TB is 5–10% [3], this period could be considered as grossly inadequate. However, another plausible reason for lower than expected TB incidence in our patients could the fact that most (82%) of them were females. In a large trial in south India [35], risk of culture positive TB in males (5%) was 3 fold higher than females (1.6%) over a follow-up period of 15 years. Comparable incidences of TB in TST-positive and TST-negative groups raise concerns about sensitivity of the methods in use for the detection of LTBI. In an earlier study, also from India, comparable proportions of TB cases emerged from TB contacts who were either positive or negative for TST or IGRA [49]. Sensitivity of these assays is also being debated on a wider scale [3, 48] and the need for more sensitive assays for a reliable detection of LTBI cannot be overemphasised [28, 31].

Mycobacteria can drive IMIDs in many ways. While the persistently high T cell responses to Mtb antigens seen by us in a subset of RA patients with high disease activity signify an adaptive immune response, part of it could also have resulted from the 'bystander' activation of T cells [12]. Activation of PRRs on innate immune cells by the pathogens can lead to bystander activation of autoreactive TH1 or TH17 cells through production of T cell promoting cytokines [50]. Moreover, T cells also express TLRs and corresponding agonists (e.g., mycobacterial antigens) can induce the proliferation of effector as well as regulatory T cells [51]. Molecular mimicry between mycobacterial and host antigens is another important pathogenic

mechanism. An Mtb-specific arthritogenic T cell clone was found to react with cartilage proteoglycan [52]. Likewise, the mycobacterial 65 kDa heat-shock protein has shown cross-reactivity with synovial tissue [53]. RA patients harbour antibodies to a peptide epitope of MAP_4027 (this antigen is also present in Mtb) which cross-reacts with interferon regulatory factor 5 (IRF5). This structural mimicry could contribute to the pathogenesis of IMIDs by dysregulating the signalling by IRF5 which is a central regulator of inflammatory responses [43, 54]. The regions of low TB prevalence (e.g., USA, Canada and Europe) show a high prevalence of MAP which could substitute for Mtb as an environmental trigger for IMIDs [55].

A major limitation of this study was less than desirable compliance of patients to the follow-up schedules. Ours is a tertiary care referral hospital where patients come from distant places. Typically, they are given medicines for 3 months and asked to return for next round of consultations. However, patients also have the option of a telephonic consultation. To partially offset this limitation, and also of a small sample size, we chose to apply a stringent cut-off ($>2$ fold increase over baseline) for classification of a patient as high responder.

To conclude, our study suggests that RA patients with LTBI may fall into distinct categories showing a persistently high or persistently low T cell response to Mtb antigens; and those showing a high response may also exhibit higher disease activity. The precise mechanisms by which preclinical RA is transformed into a clinically classifiable disease are not yet known [1, 6] and the possibility that Mtb may play a role in this transition is very much alive [56]. This possibility needs to be considered and explored, particularly in patients who are residents of a highly TB endemic area such as India. Since RA develops gradually through preclinical stages, it can be targeted much before the joint inflammation or damage sets in. There are instances when the use of anti-TB drugs has improved the outcome in RA patients [57].

## Supporting information

**S1 File.**
(PDF)

## Acknowledgments

SKK was Senior Research Fellow and SS was Emeritus Scientist of Indian Council of Medical Research, New Delhi. Excellent technical assistance was provided by Komal Singh. The work received intramural support from the host Institute (SGPGIMS).

## Author Contributions

**Conceptualization:** Ramnath Misra, Amita Aggarwal, Sudhir Sinha.

**Data curation:** Shashi Kant Kumar, Suvrat Arya.

**Formal analysis:** Shashi Kant Kumar, Suvrat Arya, Sudhir Sinha.

**Funding acquisition:** Ramnath Misra.

**Investigation:** Shashi Kant Kumar, Suvrat Arya, Sudhir Sinha.

**Methodology:** Shashi Kant Kumar, Suvrat Arya, Ankita Singh.

**Project administration:** Ramnath Misra, Amita Aggarwal, Sudhir Sinha.

**Resources:** Ramnath Misra, Amita Aggarwal.

**Supervision:** Ramnath Misra, Amita Aggarwal, Sudhir Sinha.

**Validation:** Amita Aggarwal.

**Visualization:** Sudhir Sinha.

**Writing – original draft:** Sudhir Sinha.

**Writing – review & editing:** Amita Aggarwal.

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
