## [Decision Letter · Decision Letter 0]

2 Jul 2021

PONE-D-21-17868

Patterns of T and B cell responses to Mycobacterium tuberculosis membrane-associated antigens and their relationship with disease activity in rheumatoid arthritis patients with latent tuberculosis infection

PLOS ONE

Dear Dr. Sinha,

Thank you for submitting your manuscript to PLOS ONE. After careful consideration, we feel that it has merit but does not fully meet PLOS ONE’s publication criteria as it currently stands. Therefore, we invite you to submit a revised version of the manuscript that addresses the points raised during the review process.

We look forward to receiving your revised manuscript.

Kind regards,

Leonardo A. Sechi, PhD

Academic Editor

PLOS ONE

Journal Requirements:

"The work was partly supported by an intramural grant (to RM) by the host

Institute (SGPGIMS)."

"The authors received no specific funding for this work"

Additional Editor Comments (if provided):

I've personally reviewed the manuscript and I need that you respond to the following point:

The TST response may be positive due to other mycobacterial infection, do the authors have taken in consideration to perform IFN gamma assay in order to be more specific detection MTB latent Infection?

As the 1 reviewer suggested, do other environmental or opportunistic mycobacteria (causing chronic infections) trigger RA as well?

Thank you

Leonardo A. Sechi

Reviewers' comments:

Reviewer's Responses to Questions

**Comments to the Author**

1. Is the manuscript technically sound, and do the data support the conclusions?

Reviewer #1: Yes

2. Has the statistical analysis been performed appropriately and rigorously? 

Reviewer #1: Yes

3. Have the authors made all data underlying the findings in their manuscript fully available?

Reviewer #1: Yes

4. Is the manuscript presented in an intelligible fashion and written in standard English?

Reviewer #1: Yes

5. Review Comments to the Author

Reviewer #1: In this manuscript, Shashi Kant Kumar, et al investigated the immune response to Mtb membrane antigens in RA patients with LTBI. The study is interesting and as well the results also.

However, the following are some concerns:

Different microbial agents are involved as triggers factors in RA, and TLR signalling is extremely important in host defence. In order to complete the immunopathology of RA and to better understand the molecular pathways triggered by Tb infection, the authors are invited to explain in more detail this signalling. In addition, recent in vivo studies identified the importance of Interferon Regulatory Factor 5 (IRF5) in the pathogenesis of RA as a new link between the pathogenic activation of RNA-sensing Toll-like receptors and proinflammatory cytokine production in inflamed joints of arthritic mice. Other than that, in human studies has been found a strong immune response against IRF5 suggesting IRF5 as a target of immune response in RA.

Some articles below are suggested that will certainly give more strength to the work enriching the content.

- Almuttaqi H et al. Advances and challenges in targeting IRF5, a key regulator of inflammation. FEBS J. 2019 May;286(9):1624-1637. doi: 10.1111/febs.14654. Epub 2018 Sep 21.

- Bo M et al. Antibody response to homologous epitopes of Epstein-Barr virus, Mycobacterium avium subsp. paratuberculosis (MAP) and IRF5 in patients with different connective tissue diseases and in mouse model of antigen-induced arthritis. J Transl Autoimmun. 2020 Mar 17;3:100048. doi: 10.1016/j.jtauto.2020.100048. eCollection 2020.

- Duffau P et al. Promotion of Inflammatory Arthritis by Interferon Regulatory Factor 5 in a Mouse Model. Arthritis Rheumatol. 2015 Dec;67(12):3146-57. doi: 10.1002/art.39321.

- Bo M et al. Interferon regulatory factor 5 is a potential target of autoimmune response triggered by Epstein-barr virus and Mycobacterium avium subsp. paratuberculosis in rheumatoid arthritis: investigating a mechanism of molecular mimicry. Clin Exp Rheumatol. 2018 May-Jun;36(3):376-381. Epub 2018 Jan 15.

Also, the authors have mentioned the role played by others Mycobacteria in RA, and appears interesting the role played by Mycobacterium avium subsp. paratuberculosis as a possible trigger factor recently discovered in the aetiology of RA. It would better give more details writing a short and concise paragraph on the role of Mycobacterial infections in RA.

Finally, a recent review on the role of Mycobacterial infections in RA is suggested below in order to complete your manuscript.

- Bo M et al. Role of Infections in the Pathogenesis of Rheumatoid Arthritis: Focus on Mycobacteria. Microorganisms. 2020 Sep 23;8(10):1459. doi: 10.3390/microorganisms8101459.

Best wishes

6. PLOS authors have the option to publish the peer review history of their article (what does this mean?). If published, this will include your full peer review and any attached files.

Reviewer #1: No

---

## [Author Response · Author response to Decision Letter 0]

12 Jul 2021

Response to Academic Editor

1. The TST response may be positive due to other mycobacterial infection, do the authors have taken in consideration to perform IFN gamma assay in order to be more specific detection MTB latent Infection?

Response: Going by the available literature (Ref 31, 38, 39 of the manuscript) and WHO recommendations (Ref 3 of the manuscript), TST results are least likely to be affected by BCG vaccination (if given in infancy) or NTM infection. We have further elaborated this point in the revised Discussion (paragraph 2).

2. As the 1 reviewer suggested, do other environmental or opportunistic mycobacteria (causing chronic infections) trigger RA as well?

Response: Yes, they do. We have made necessary amendments in the Introduction (para 2) and Discussion (para 3, 5 and 8).

Response to Reviewer-1

1. Different microbial agents are involved as triggers factors in RA, and TLR signalling is extremely important in host defence. In order to complete the immunopathology of RA and to better understand the molecular pathways triggered by Tb infection, the authors are invited to explain in more detail this signalling. In addition, recent in vivo studies identified the importance of Interferon Regulatory Factor 5 (IRF5) in the pathogenesis of RA as a new link between the pathogenic activation of RNA-sensing Toll-like receptors and proinflammatory cytokine production in inflamed joints of arthritic mice. Other than that, in human studies has been found a strong immune response against IRF5 suggesting IRF5 as a target of immune response in RA.

Response: Thanks for providing this insight. We have made necessary amendments in the Introduction (para 2) and inserted a new paragraph (para 8) in the Discussion which hopefully addresses these aspects.

2. Some articles below are suggested that will certainly give more strength to the work enriching the content.

(i) Almuttaqi H et al. Advances and challenges in targeting IRF5, a key regulator of inflammation. FEBS J. 2019 May;286(9):1624-1637. doi: 10.1111/febs.14654. Epub 2018 Sep 21.

(ii) Bo M et al. Antibody response to homologous epitopes of Epstein-Barr virus, Mycobacterium avium subsp. paratuberculosis (MAP) and IRF5 in patients with different connective tissue diseases and in mouse model of antigen-induced arthritis. J Transl Autoimmun. 2020 Mar 17;3:100048. doi: 10.1016/j.jtauto.2020.100048. eCollection 2020.

(iii) Duffau P et al. Promotion of Inflammatory Arthritis by Interferon Regulatory Factor 5 in a Mouse Model. Arthritis Rheumatol. 2015 Dec;67(12):3146-57. doi: 10.1002/art.39321.

(iv) Bo M et al. Interferon regulatory factor 5 is a potential target of autoimmune response triggered by Epstein-barr virus and Mycobacterium avium subsp. paratuberculosis in rheumatoid arthritis: investigating a mechanism of molecular mimicry. Clin Exp Rheumatol. 2018 May-Jun;36(3):376-381. Epub 2018 Jan 15.

Response: We have incorporated references (ii) and (iv) in the revised Introduction (para 2) and Discussion (para 3, 5 and 8). However, we did not include (i) and (iii) since we felt that ref (ii), (iv) and another one suggested by you (see below, point 4) cover the necessary information. All along, we were wary of exceeding our brief since our present work does not address these aspects.

3. Also, the authors have mentioned the role played by others Mycobacteria in RA, and appears interesting the role played by Mycobacterium avium subsp. paratuberculosis as a possible trigger factor recently discovered in the aetiology of RA. It would better give more details writing a short and concise paragraph on the role of Mycobacterial infections in RA.

Response: Thanks for the suggestion. We have added a new paragraph 8 in the revised Discussion.

4. Finally, a recent review on the role of Mycobacterial infections in RA is suggested below in order to complete your manuscript.

Bo M et al. Role of Infections in the Pathogenesis of Rheumatoid Arthritis: Focus on Mycobacteria. Microorganisms. 2020 Sep 23;8(10):1459. doi: 10.3390/microorganisms8101459.

Response: We have now incorporated this reference (Introduction para 1).

---

## [Decision Letter · Decision Letter 1]

21 Jul 2021

Patterns of T and B cell responses to Mycobacterium tuberculosis membrane-associated antigens and their relationship with disease activity in rheumatoid arthritis patients with latent tuberculosis infection

PONE-D-21-17868R1

Dear Dr. Sudhir Sinha

We’re pleased to inform you that your manuscript has been judged scientifically suitable for publication and will be formally accepted for publication once it meets all outstanding technical requirements.

Kind regards,

Leonardo A. Sechi, PhD

Academic Editor

PLOS ONE

Additional Editor Comments (optional):

Reviewers' comments:

Reviewer's Responses to Questions

**Comments to the Author**

1. If the authors have adequately addressed your comments raised in a previous round of review and you feel that this manuscript is now acceptable for publication, you may indicate that here to bypass the “Comments to the Author” section, enter your conflict of interest statement in the “Confidential to Editor” section, and submit your "Accept" recommendation.

Reviewer #1: All comments have been addressed

2. Is the manuscript technically sound, and do the data support the conclusions?

Reviewer #1: Yes

3. Has the statistical analysis been performed appropriately and rigorously? 

Reviewer #1: Yes

4. Have the authors made all data underlying the findings in their manuscript fully available?

Reviewer #1: Yes

5. Is the manuscript presented in an intelligible fashion and written in standard English?

Reviewer #1: Yes

6. Review Comments to the Author

Reviewer #1: The authors have adequately addressed comments raised by reviewers and I feel that this manuscript is now acceptable for publication.

7. PLOS authors have the option to publish the peer review history of their article (what does this mean?). If published, this will include your full peer review and any attached files.

Reviewer #1: No

---

## [Editor Report · Acceptance letter]

23 Jul 2021

PONE-D-21-17868R1 

**Patterns of T and B cell responses to *Mycobacterium tuberculosis* membrane-associated antigens and their relationship with disease activity in rheumatoid arthritis patients with latent tuberculosis infection**

Dear Dr. Sinha:

I'm pleased to inform you that your manuscript has been deemed suitable for publication in PLOS ONE. Congratulations! Your manuscript is now with our production department. 

Kind regards, 

on behalf of

Professor Leonardo A. Sechi 

Academic Editor

PLOS ONE